# Calcinosis in Rheumatic Disease Is Still an Unmet Need: A Retrospective Single-Center Study

**DOI:** 10.3390/diagnostics14060637

**Published:** 2024-03-18

**Authors:** Cristina Nita, Laura Groseanu, Daniela Opris, Denisa Predeteanu, Violeta Bojinca, Florian Berghea, Violeta Vlad, Mihai Abobului, Cosmin Constantinescu, Magdalena Negru, Ioana Saulescu, Sanziana Daia, Diana Mazilu, Andreea Borangiu, Claudia Cobilinschi, Denisse Mardale, Madalina Rosu, Andra Balanescu

**Affiliations:** 1Department of Internal Medicine and Rheumatology, ‘Sfanta Maria’ Clinical Hospital, 011172 Bucharest, Romania; elena-cristina.nita@drd.umfcd.ro (C.N.); daniela.opris@umfcd.ro (D.O.); florian.berghea@umfcd.ro (F.B.); denise-ani.mardale@drd.umfcd.ro (D.M.);; 2Faculty of Medicine, University of Medicine and Pharmacy ‘Carol Davila’, 050474 Bucharest, Romania

**Keywords:** calcinosis, immune-mediated rheumatic diseases, patient care, outcomes

## Abstract

Patients with immune-mediated rheumatic disease-related calcinosis comprise a subgroup at risk of encountering a more severe clinical outcome. Early assessment is pivotal for preventing overall disease progression, as calcinosis is commonly overlooked until several years into the disease and is considered as a ‘non-lethal’ manifestation. This single-center retrospective study explored the prevalence, clinical associations, and impact on survival of subcutaneous calcinosis in 86 patients with immune-mediated rheumatic diseases (IMRD). Calcinosis predominantly appeared in individuals with longstanding disease, particularly systemic sclerosis (SSc), constituting 74% of cases. Smaller calcinosis lesions (≤1 cm) were associated with interstitial lung disease, musculoskeletal involvement, and digital ulcerations, while larger lesions (≥4 cm) were associated with malignancy, severe peripheral artery disease, and systemic arterial hypertension. The SSc calcinosis subgroup exhibited a higher mean adjusted European Scleroderma Study Group Activity Index score than those without. However, survival rates did not significantly differ between the two groups. Diltiazem was the most commonly used treatment, and while bisphosphonates reduced complications related to calcinosis, complete resolution was not achieved. The findings underscore current limitations in diagnosing, monitoring, and treating calcinosis, emphasizing the need for further research and improved therapeutic strategies to improve patient care and outcomes.

## 1. Introduction

Soft-tissue calcifications have been classified into five major types: dystrophic, metastatic, idiopathic, iatrogenic, and calciphylaxis [1]. In immune-mediated rheumatic diseases (IMRD), dystrophic calcification, characterized by the accumulation of calcium salts in areas of persistent inflammation or previous injury, with normal serum calcium and phosphorus levels, is common [2]. The etiology of calcinosis remains uncertain, with proposed mechanisms including microvascular changes [3,4,5], the dysregulation of components of the innate immune system [6], and associations with specific autoantibodies (e.g., anti-MDA5, anti-NXP2, anti-PM/Scl, anti-centromere, and anti-topoisomerase I) [7]. Additional theories postulate contributions from recurrent trauma and abnormalities in bone matrix proteins [4,8]. The prevalence of calcinosis is often underestimated, leading to delayed diagnosis and long-term complications. It may vary depending on the subset of the disease: 20–40% in systemic sclerosis (SSc) patients, with 25–40% in limited cutaneous SSc [9,10]; 17% in systemic lupus erythematosus (SLE) [11]; and 30–70% in dermatomyositis (DM), particularly in childhood-onset cases [12,13]. Additionally, calcinosis, though rare, has been reported in polymyositis (PM), rheumatoid arthritis (RA), localized scleroderma (i.e., morphea), mixed connective disease (MCTD), and primary Sjögren syndrome (pSS) [9,14]. Common clinical symptoms of calcinosis include pain, recurrent local inflammation, ulcers with infections, and deformities, causing functional impairment. Calcinosis can also lead to compression neuropathies, with motor and sensory deficits, with a higher likelihood of skin ulceration in areas like the forearm, elbows, fingers (especially the volar aspect of fingertips), and metacarpophalangeal and interphalangeal joints [15]. Radiological investigations can contribute to early diagnosis, treatment guidance, and prognosis assessment [16]. The Scleroderma Clinical Trials Consortium (SCTC) has developed a validated radiographic scoring system for hand SSc-related calcinosis, considering factors like area, density, number, and location to estimate calcinosis burden [17]. Currently, the absence of established guidelines and standardized tools hinders the evaluation of pharmacologic treatments for calcinosis in IMRD. Additionally, there are no established patient-reported outcome measures (PROMs) for IMRD-related calcinosis, except for the SCTC Mawdsley Calcinosis Questionnaire, which is currently undergoing validation [18]. Despite this therapeutic challenge, promising emerging approaches include the systemic administration of sodium thiosulfate or immunosuppressant agents [19]. Surgery may be the only treatment option in some cases, but outcomes are unpredictable [20]. Demographic characteristics and clinical features of individuals with IMRD-related calcinosis have been scarcely documented, with a focus on SSc and myositis patients. Consequently, a notable deficiency exists in comprehensive data regarding the clinical course, long-term follow-up, associations with other disease-specific characteristics, and efficacy of treatment options for this condition. Our ongoing research, conducted at a tertiary-care rheumatology unit with access to a multidisciplinary committee, aims to fill this gap by identifying clinical and immunological prognostic factors and evaluating practical treatment approaches for IMRD-related calcinosis. 

## 2. Materials and Methods

### 2.1. Study Design

This work is a retrospective cross-sectional observational study and was conducted in the Rheumatology Department of “Saint Mary’s” Clinical Hospital, in Bucharest, Romania, between January 2000 and December 2022. Data from 86 selected patients were collected retrospectively from hospital charts. This study was conducted according to the guidelines of the Declaration of Helsinki and was approved by the Ethics Committee for Research of the “Carol Davila” University of Medicine and Pharmacy, Bucharest. Informed consent was waived due to the retrospective nature of the study.

Patient selection cases were initially selected from the computerized database using keywords associated with the names of specific IMRDs (e.g., “systemic sclerosis”, “dermatomyositis”, “polymyositis”, “systemic lupus erythematosus”, “primary Sjögren syndrome”, “mixed connective tissue disease”, and “overlap connective tissue disease”). The initial search yielded 872 patients. Subsequently, the medical records of these patients were scrutinized using the terms “calcinosis”, “calcifications”, and “dystrophic calcification”, leading to the identification of 86 patients meeting the inclusion criteria for IMRD- related calcinosis. Each IMRD patient satisfied their respective disorder’s classification criteria [21,22,23,24,25]. Patients with incidentally identified forms of calcinosis unrelated to IMRDs were excluded. 

### 2.2. Data Collection

Demographic data including age at onset, gender, ethnicity, IMRD diagnosis, follow-up duration, disease duration, anatomic distribution, and clinical presentation were systematically collected and documented. Locations of calcinosis were categorized as follows: extremities (including elbows, knees, shoulders, buttocks, and pretibial area prone to friction or mild trauma), hands/fingers, feet, and atypical areas (head, spine, trunk, iliac crest, and intergluteal area). The adjusted European Scleroderma Study Group Activity Index (EScSG AI) was computed for all 64 SSc patients. Radiologic images including plain radiographs, ultrasonographic images, computed tomography scans, and/or magnetic resonance imaging (MRI) were also compiled. Additionally, we conducted a comprehensive examination medical chart to rule out soft tissue calcifications related to metabolic factors. Apart from routine hematologic, biochemical, and metabolic tests, we retrieved immunological profiles from patients’ charts. Antinuclear antibodies (ANA) were tested by indirect immunofluorescence (IIF) and considered positive at titer ≥ 1:160. Autoantibodies against various specific antigens, including topoisomerase I (anti-Scl70), anti-Smith (anti-Sm), anti-ribonucleic protein antigen (anti-Sm/RNP), anti-cyclic citrullinated peptide (CCP), anti-double-stranded deoxyribonucleic acid (dsDNA), Sjögren’s syndrome-related antigens (SSA, SSB), anti-U1 RNP, and antiphospholipid antibodies, were determined using enzyme-linked immunosorbent assay (ELISA). Plasma was also screened for DM-specific and associated autoantibodies like anti-nuclear matrix protein 2 (anti-NXP-2), anti-melanoma differentiation-associated protein 5 (anti-MDA-5), anti-transcriptional intermediary factor 1 gamma (anti-TIF1-γ), anti-Mi-2 gamma (anti-Mi-2), anti-Ku complex (anti-Ku), anti-polymyositis/scleroderma (anti-Pm/Scl), anti-small ubiquitin-like modifier protein conjugation system (anti-SAE1), and anti-histidyl-tRNA synthetase antibodies (anti-Jo-1) (Alegria^TM^, Orgentec, Germany). 

### 2.3. Treatment and Outcomes

We documented a range of therapeutic approaches, including medications, surgeries, and complementary treatments. We also conducted a thorough evaluation of treatment outcomes, assessing factors such as wound healing, changes in lesion size and number, and the occurrence of complications. Patients’ responses to therapy were categorized into four grades: complete, partial, none, or unknown. Complete response denoted the total resolution of an individual lesion without subsequent recurrence in the same area. The partial response indicated regression or recurrence of a lesion that had previously regressed or completely healed. No response was attributed to the persistence of old lesions, with or without the emergence of new ones. 

### 2.4. Data Analysis 

For analysis, individual cases were paired with controls based on age and gender. Our cohort comprised 86 patients with IMRD and calcinosis, while 90 IMRD patients without calcinosis were selected as the control group. Descriptive statistics, such as median values, standard deviations (SD), or percentages, were employed to summarize data. Clinical and immunological variables, with potential predictive value for calcinosis development, were entered into a logistic regression analysis. Comparisons between groups, stratified by the topographic distribution of calcinosis, were conducted using Fisher’s exact test or a Wilcoxon ranked test, as appropriate. 

## 3. Results 

### 3.1. Patient Characteristics

#### 3.1.1. General Description of the Study Lot 

Within the cohort, calcinosis related to IMRD was observed in 86 individuals, representing 9.7% of total cases. The calcinosis group consisted of 71 females and 15 males, with a mean age of 52.6 years (±14.3) and an average disease duration of 5.6 years (±3.1). The time from disease onset to calcinosis development varied significantly among different IMRDs, with a notably shorter duration in diffuse cutaneous SSc (dcSSc) compared to Sjögren’s syndrome (1.1 vs. 21.5 years, *p* < 0.001). The distribution of cases and patients’ characteristics are summarized in Table 1. Among the 8 patients diagnosed with scleroderma overlap-related calcinosis, limited cutaneous SSc (lcSSc) and PM emerged as the prevailing disease phenotypes, with additional associations observed with RA, DM, pSS, and psoriatic arthritis. 

#### 3.1.2. Anatomic Distribution of Calcinosis

All clinically suspected calcifications were confirmed through various imaging studies (radiography, computed tomography, MRI, or ultrasound). Skin biopsy was used to confirm the diagnosis in only two patients. While in patients with lcSSc, the hands (96%) were the most frequently affected sites, followed by areas exposed to frequent pressure such as the elbows and knees (76%), in dcSSc patients, the distribution was more widespread. Patients with PM/DM and overlap syndromes often presented with multiple affected sites at disease onset, notably in the pelvic girdle, lower limbs, and extremities. Furthermore, 73% of these patients had calcifications in less common areas such as the maxillary sinuses, cervical spine, chest, and buttocks, often with nonspecific symptoms. In SLE patients, the upper limbs and abdominal wall were the most commonly affected areas (Figure 1). 

### 3.2. Correlation Analyses between Different IMRD and Subcutaneous Calcinosis

The analyses primarily centered on SSc patients, given their predominance within the study cohort. Assessing this aspect for other IMRD subsets posed challenges due to limited sample sizes and heterogeneous presentations. Factors correlated with calcinosis in all the SSc patients encompassed disease duration, female gender, digital ulcers (DU), and severe GI involvement (Table 2). 

In the SSc subgroup, univariate logistic regression analysis showed that lesions ≤ 1 cm were associated with anti-topoisomerase antibodies, interstitial lung disease, articular involvement, and DU, while those ≥4 cm were linked to anti-Pm/Scl antibodies, symptoms of muscle weakness, severe peripheral artery disease, systemic arterial hypertension, malignancy, and higher disease activity reflected by the EScSG-AI (Table 3). Moreover, PM/Scl antibodies were correlated with longer disease duration [OR 4.50 95% CI 1.39–14.6, *p* = 0.01], whereas anti-TIF1-γ was associated with a lower calcinosis prevalence in adult DM [OR 0.80, 95% CI 0.85–0.99, *p* = 0.03].

### 3.3. Treatment Strategies and Outcomes in Patients with IMRD—Related Calcinosis

#### 3.3.1. Treatment of Calcinosis Cutis 

All IMRD-related calcinosis patients received various treatments, including calcium channel blockers such as diltiazem (18/86 patients, 120 to 480 mg/day) or nifedipine (8/86 patients, 20 mg/day), bisphosphonates (alendronic acid in 11% of patients, ibandronic acid in 7% of patients, risedronic acid and zoledronate in 2% of patients), as well as colchicine and minocycline (Figure 2). Warfarin sodium or intravenous immunoglobulin were not prescribed, and neither were topical or intralesional steroids, nor sodium thiosulfate. Surgical excision was performed in only one patient due to pain and vascular compression caused by a massive calcification on the index finger. Diltiazem was the most frequently used first-line therapy for calcinosis, with a median duration of 68 (12–240) months. Combining colchicine (1 mg/day) with diltiazem for calcinosis-related complications resulted in a partial response in 16 out of 27 patients over an average of a 28 month duration. Additionally, among 15 patients with painful or ulcerated nodules who were prescribed minocycline (50–100 mg daily), 12 showed a partial response. The earliest improvement was observed after 2 weeks, with an average resolution time of 2.4 ± 2.8 months. The further analysis of data on immunomodulatory agent use, including type and number, for all IMRD-related calcinosis patients revealed no significant correlation between the extent of calcinosis-affected areas and the number of previous immunomodulatory agents used. 

#### 3.3.2. Outcomes and Survival 

A proportion of 58% (49/86) of patients experienced calcinosis-related complications, with around 42% of requiring hospitalization for infections and wound care debridement. Of these, 27 individuals (26 with SSc and 1 with DM) had ≥4 admissions per year. Common complications included tenderness (53%), functional joint impairment (42%), and fistulas (37%). Patients with calcinosis in the finger/thumbs had a higher incidence of complications compared to those with calcinosis elsewhere (86.36% vs. 51.57%, *p* = 0.002). In specific IMRD subsets, such as SSc, patients with Raynaud’s phenomenon, DU, and GI involvement, were more prone to recurrent complications related to calcinosis (Table 4). In DM and scleroderma overlap PM, subcutaneous tumoral deposits led to joint limitations and secondary infections, while SLE patients experienced small calcifications causing localized pain without significant inflammation. Nine patients passed away, primarily due to causes such as respiratory failure (3 patients), ventricular arrhythmias (2 patients), and scleroderma renal crisis (1 patient). The presence of calcinosis did not significantly affect survival rates in the study cohort or the SSc subset, as evidenced by no difference in median survival time between those with and without calcinosis (9.92 ± 7.5 vs. 10.5 ± 9.7, *p* = 0.22).

## 4. Discussion

In our study involving 872 patients with immune-mediated rheumatic diseases, calcinosis was observed in 86 individuals (9.7%), aligning with a previously reported prevalence ranging from 8.45% to 19.54% [2,15,26]. Calcinosis was more common in SSc, affecting 20% of patients (64/310). Similar to Mexican and Canadian cohorts [3,27], our patient population mainly had diffuse cutaneous systemic sclerosis, accounting for 59.37%. The most commonly involved sites were the hands and feet, especially the fingers, consistent with findings in prior studies [15,28]. Additionally, SSc patients often exhibited ulcers (55%), impacting 81% of dcSSc and 56% of lcSSc patients. Consistent with Bartoli et al. [29], superimposed infections occurred more frequently, affecting 62% of cases. In our myositis subgroup (n = 146), calcinosis prevalence also aligned with recent reports (14%) [12,30]. All eight overlap syndrome patients exhibited scleroderma overlap syndromes, mostly lcSSc. Although minimal calcinosis is acknowledged in overlap syndromes, documented cases remain rare [28,31,32]. Six SLE patients had calcinosis and one had lupus panniculitis. Despite Okada’s report of a high prevalence of ectopic calcifications (40%) in SLE patients, particularly in periarticular areas (33%) and soft tissues (17%) [33], it is noteworthy that dystrophic calcinosis remains infrequent in SLE, with only 45 documented cases in the literature [1,9].

Factors associated with a higher prevalence of calcinosis in the SSc subgroup included female gender, DU, disease duration, and severe GI involvement. Previous studies have also reported a higher likelihood of calcinosis development in SSc patients with longer disease duration, typically around 7–10 years [34]. In contrast, despite Muktabhant et al. suggesting a potential higher risk of early calcinosis development in elderly patients [35], our cohort did not show age-related differences. The association between DU and calcinosis aligns with prior research indicating a higher likelihood of calcinosis in patients with severe vascular digital involvement [3]. An international cohort study identified additional risk factors, including osteoporosis, and reduced carbon monoxide diffusion capacity/alveolar volume [36]. Recent findings by Richardson et al. demonstrated that calcinosis burden correlates with cumulative SSc-related tissue damage, irrespective of disease severity. Patients with predominantly diffuse skin disease and more severe disease features, including pulmonary hypertension, interstitial lung disease, cardiomyopathy, gastrointestinal involvement, renal crisis, myopathy, and/or tendon friction rubs, had an elevated risk of calcinosis compared to those with limited skin disease [37]. Furthermore, in SSc patients, calcinosis has been linked to exposure to proton pump inhibitors (PPIs). Prolonged PPI treatment, exceeding 10 years, increased the likelihood of calcinosis at any time seven-fold and raised the odds of calcinosis cutis nearly eight-fold when compared to no PPI exposure [38]. While we could not identify a definitive clinical profile of risk factors for calcinosis in our myositis patients, Fredi et al. considered anti-NXP-2 antibody as an independent predictive risk factor for calcinosis development (*p* = 0.024, OR 21.9, CI 95% = 1.5–319) [39]. Moreover, the authors suggested that anti-NXP-2 antibody positivity represents a distinct calcinosis phenotype with early onset and rapid dissemination. Two of our patients with calcinosis tested positive for anti-NXP-2 antibodies, and both were diagnosed with DM and exhibited widespread involvement from disease onset. Moreover, prior research has shown that PM/Scl antibody positivity, prolonged muscle enzyme elevation, and the use of multiple immunosuppressive therapies are associated with calcinosis development in myositis patients [40]. The connection between calcinosis and anti-PM/Scl antibodies was most relevant in our cohort, and this association has been previously documented in the literature [8,41]. Okada and colleagues [33] noted that alfacalcidol therapy and lupus nephritis might increase the risk of calcinosis in SLE patients. We compared alfacalcidol therapy between the calcinosis-positive and negative groups and found no significant differences. We observed only two cases of nephritis and there were no apparent abnormalities in the progression of calcinotic lesions. Another recent research study of particular interest focused on the disease course of ten lupus-related calcinosis patients. Among these patients, eight reported Raynaud’s, and seven of them were anti-RNP positive. The authors suggested that lupus patients who exhibit some features resembling MCTD may be more prone to developing calcinosis [11]. 

In terms of internal organ involvement, the analyses centered on the SSc patient cohort, as most calcinosis cases were in this group. Univariate analysis revealed that calcinosis patients exhibited a higher proportion of severe organ involvement, particularly in vascular and GI disease. Other clinical features did not significantly correlate with the calcinosis subset in SSc, although recent research suggests a positive link between subcutaneous calcinosis at disease onset and the presence of pulmonary arterial hypertension [42]. SSc patients had a higher frequency of DU history, which aligns with the proposal of a link between calcinosis and vasculopathy by Avouac et al., as their patient population with calcinosis was more likely to have a history of DU [4]. Masatoshi et al. classified calcinosis as a “broad-sense” vascular lesion, proposing topical blood flow disturbance and vascular hypoxia as etiological factors [43]. This hypothesis finds support in a pilot study demonstrating reduced perfusion in the superficial skin layers of calcinotic areas compared to non-calcinotic areas [44]. Despite some controversy, the strongest associations with calcinosis in the univariate analysis were severe peripheral artery disease and systemic arterial hypertension. These findings are novel, lacking previous systematic investigation, though Lescoat et al. have reported an association between calcinosis and ulnar artery occlusion [45]. However, clinical parameters like a history of other cardiovascular risk factors (smoking, dyslipidemia, or hyperglycemia) did not show associations with calcinosis features in univariable analyses. 

## 5. Conclusions

In summary, our data have revealed various clinical associations with calcinosis across different patient populations. However, larger prospective studies are needed to confirm these preliminary findings. A definitive clinical profile of risk factors for calcinosis in IMRD patients remains elusive, with current information primarily derived from retrospective patient series. While the retrospective nature of this study may limit result interpretation, it is one of the largest studies of IMRD-related calcinosis in the literature. Further research is imperative to enhance our understanding and management of this challenging condition, which currently lacks a standardized approach and may vary in treatment outcomes based on multiple factors.

## Figures and Tables

**Figure 1 diagnostics-14-00637-f001:**
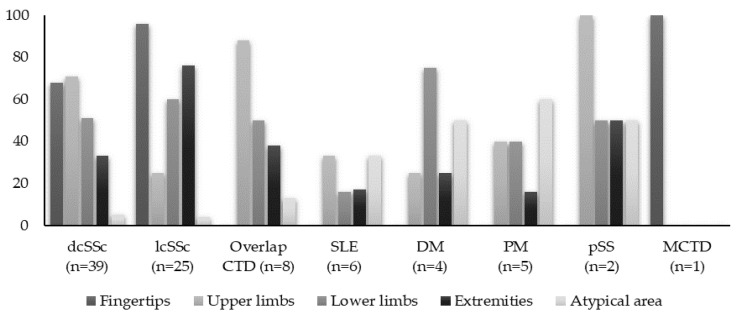
The anatomical distribution of calcinotic lesions across the study cohort. The extremities encompassed the elbows, knees, shoulders, buttocks, and pretibial area. Atypical areas included the spine, trunk, iliac crest, and intergluteal area.

**Figure 2 diagnostics-14-00637-f002:**
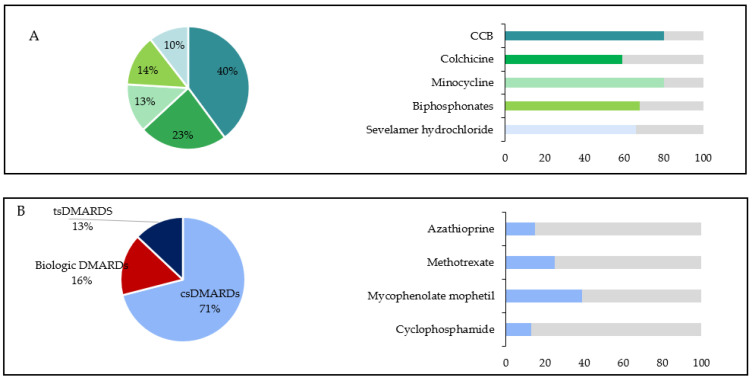
Treatment approaches for 86 patients with IMRD-related calcinosis. The treatment modalities are presented in a pie chart showing the overall distribution, while the treatment response is depicted in bar charts. The *x*-axis represents the percentage of patients with a partial response, indicated by regression of a lesion, and the *y*-axis represents the type of treatment received: (**A**). Pharmacologic treatment: calcium channel blockers (CCB, 40% of patients), colchicine (23%), minocycline (13%), bisphosphonates (14%), sevelamer hydrochloride (10%); (**B**). Immunomodulatory treatment: conventional synthetic disease-modifying antirheumatic drugs (csDMARDs): azathioprine (13%), methotrexate (25%), mycophenolate (39%), and intravenous pulse cyclophosphamide (13%); biologic DMARDs: rituximab 5%, etanercept 10%; targeted synthetic DMARDs (tsDMARDs): tofacitinib (13%).

**Table 1 diagnostics-14-00637-t001:** Characteristics and prevalence of calcinosis in the studied cohort.

	dcSSc (*n* = 39)	lcSSc (*n* = 25)	Overlap CTD ^a^ (*n* = 8)	SLE (*n* = 6)	PM (*n* = 5)	DM (*n* = 4)	pSS (*n* = 2)	MCTD (*n* = 1)	*p*
Demographics									
Female	37 (49)	19 (79)	8 (100)	6 (100)	5 (100)	4 (100)	2 (100)	1 (100)	0.58
Age, (IQR), years	50 (36–60)	54 (28–73)	39 (24–55)	29 (20–48)	39 (24–55)	44 (37–68)	46 (35–67)	39	0.18
Time to onset of calcinosis, (IQR), months	13 (2–43)	90 (0–372)	128 (2–312)	24 (1–55)	108 (18–546)	48 (11–109)	258 (228–742)	75	<0.001
Total observation period, (IQR), years									0.09
Raynaud’s	31 (79)	23 (92)	8 (100)	1 (16)	-	-	-	1 (100)	0.54
Telangiectasias	18 (46)	23 (92)	7 (88)	-	-	-	-	1 (100)	0.49
Digital ulcers	20 (51)	16 (64)	5 (62)	-	-	-	-	-	0.03
Articular involvement	22 (56)	24 (96)	6 (75)	3 (50)	2 (40)	1 (25)	1 (50)	1 (100)	0.06
Muscle involvement	14 (36)	16 (64)	7 (88)	2 (33)	5 (100)	4 (100)	2 (100)	-	0.10
GIT involvement	18 (46)	21 (84)	4 (50)	1 (16)	1 (20)	1 (25)	-	1 (100)	0.003
Severe GIT disease	14 (35)	4 (16)	2 (25)	-	-	-	-	1 (100)	<0.001
Peripheric arterial disease	25 (64)	20 (80)	3 (37)	-	-	-	-	-	0.01
Severe peripheral arterial disease	19 (48)	9 (36)	3 (37)	-	-	-	-	-	0.02
ILD	26 (66)	15 (60)	5 (63)	-	2 (40)	-	-	1 (100)	0.28
PAH	13 (33)	11 (44)	2 (25)	-	-	-	-	1 (100)	0.17
Myocardial disease	14 (36)	6 (24)	1 (12)	-	-	-	-	1 (100)	0.21
Chronic kidney disease	4 (10)	2 (8)	1 (12)	1 (17)	-	-	1 (50)	-	0.15
Osteoporosis	12 (31)	9 (36)	2 (25)	1 (17)	2 (40)	-	-	-	0.81
Arterial hypertension	12 (31)	13 (52)	1 (12)	-	1 (20)	1 (25)	1 (50)	-	0.25
Diabetes	3 (8)	2 (8)	-	-	-	1 (25)	-	-	0.10
Dyslipidemia	8 (20)	8 (32)	3 (37)	-	-	1 (25)	1 (50)	-	0.11
Malignancy	4 (10)	1 (4)	1 (12)	-	1 (20)	-	-	-	0.35

Abbreviations: Data are presented as median ± interquartile range or n (%), unless otherwise stated. IMRD: immune-mediated rheumatic diseases; dcSSc: diffuse cutaneous systemic sclerosis; lcSSc: limited cutaneous systemic sclerosis; CTD: connective tissue disease; SLE: systemic lupus erythematosus; MTCD: mixed connective tissue disease; IQR: interquartile range; NS = not significant. ^a^ The 8 patients in this cohort had overlapping CTDs of the following types: SSc and rheumatoid arthritis (*n* = 3), SSc and polymyositis (*n* = 3), SSc and Sjögren syndrome (*n* = 1), and SSc and psoriatic arthritis (*n* = 1).

**Table 2 diagnostics-14-00637-t002:** Baseline characteristics associated with the presence of subcutaneous calcinosis in SSc patients.

	Multivariate Analysis	
Variable	OR (95% CI)	*p*-Value
Age	0.99 (0.97–1.01)	0.88
Female	1.83 (1.12–2.95)	**0.02**
Disease duration	1.04 (1.00–1.08)	**0.001**
ATA	1.00 (0.59–1.68)	0.98
ACA	1.42 (0.84–2.40)	0.18
Anti-Pm/Scl	7.58 (1.13–9.17)	**0.004**
mRSS	1.01 (0.97–1.05)	0.56
Tendon friction rubs	0.60 (0.22–1.66)	0.33
Synovitis	0.82 (0.42–1.60)	0.56
Muscle weakness	0.81 (0.45–1.47)	0.50
Upper GI symptoms	0.90 (0.52–1.56)	0.71
Lower GI symptoms	1.44 (0.81–2.53)	0.20
Severe GI disease	1.27 (1.02–14.76)	**0.001**
PAH	1.96 (0.93–4.41)	0.07
DU	1.36 (1.06–3.03)	**0.004**
EScSG AI	1.09 (1.01–1.46)	**0.05**

Abbreviations: ATA: anti-topoisomerase antibodies; ACA: anti-centromere antibodies; mRSS: modified Rodnan Skin Score; PAH: pulmonary arterial hypertension; DU: digital ulcers; EScSG AI: the adjusted European Scleroderma Study Group Activity Index. Statistically significant differences, with a probability value of *p* < 0.05, are represented in bold.

**Table 3 diagnostics-14-00637-t003:** Associations between dimensions of calcinosis lesions and patients’ characteristics in the SSc subgroup.

	Subcutaneous Lesions Size			
Variable	≤1 cm		≥4 cm	
	OR (95% CI)	*p*-Value	OR (95% CI)	*p*-Value
ATA	1.90 (1.06–5.57)	0.04	1.42 (0.50–4.01)	0.50
ACA	0.41 (0.14–1.19)	0.10	0.66 (0.22–1.98)	0.46
Anti-Pm/Scl	0.42 (0.13–1.28)	0.12	2.15 (1.70–6.57)	**0.01**
mRSS	1.02 (0.94–1.11)	0.51	0.98 (0.91–1.06)	0.72
Tendon friction rubs	1.06 (0.36–3.11)	0.91	2.05 (0.53–7.97)	0.29
Synovitis	1.45 (1.05–3.40)	0.02	1.00 (0.31–3.23)	0.99
Muscle weakness	0.65 (0.22–1.89)	0.43	3.12 (1.04–9.31)	**0.04**
Upper GI symptoms	0.89 (0.25–3.13)	0.86	0.83 (0.25–2.72)	0.76
Lower GI symptoms	1.72 (0.34–8.61)	0.50	1.54 (0.41–5.71)	0.51
Severe GI disease	1.43 (0.41–4.94)	0.57	0.92 (0.28–2.96)	0.89
Peripheral arterial disease	2.38 (0.61–9.17)	0.20	1.48 (0.43–5.11)	0.53
Severe peripheral arterial disease	0.55 (0.09–3.28)	0.51	1.48 (1.03–5.11)	**0.007**
Systemic arterial hypertension	1.60 (0.62–4.11)	0.32	1.83 (1.29–2.37)	**<0.001**
PAH	1.06 (0.36–3.11)	0.91	1.01 (0.98–1.04)	0.46
ILD	2.13 (1.77–6.92)	0.01	0.39 (0.13–1.17)	0.09
DU	2.44 (1.83–7.10)	0.02	0.83 (0.29–2.37)	0.73
Malignancy	1.02 (0.97–1.09)	0.34	1.01 (1.00–3.23)	**0.01**
EScSG AI	1.37 (0.46–4.04)	0.56	2.06 (1.68–6.24)	**0.02**

Abbreviations: ATA: anti-topoisomerase antibodies; ACA: anti-centromere antibodies; mRSS: modified Rodnan Skin Score; PAH: pulmonary arterial hypertension; ILD: interstitial lung disease; DU: digital ulcers; EScSG AI: the adjusted European Scleroderma Study Group Activity Index. Statistically significant differences, with a probability value of *p* < 0.05, are represented in bold.

**Table 4 diagnostics-14-00637-t004:** Binary logistic regression regarding independent predictors of calcinosis-related complications in the entire cohort.

	Multivariate Analysis	
Variable	OR (95% CI)	*p* Value
Raynaud’s phenomenon	6.79 (2.63–9.31)	0.06
DU	3.68 (1.39–9.74)	**0.009**
Severe GI disease	2.95 (1.88–9.86)	**0.01**

Abbreviations: DU: digital ulcers. Statistically significant differences, with a probability value of *p* < 0.05, are represented in bold.

## Data Availability

Data can be made available upon reasonable request due to ethical restrictions.

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
