# Peer review of "Calcinosis in Rheumatic Disease Is Still an Unmet Need: A Retrospective Single-Center Study"

_diagnostics, 2024, doi:10.3390/diagnostics14060637_

Round 1
Reviewer 1 Report
Comments and Suggestions for Authors
The authors present an observational study that aims to explored the prevalence, clinical associations, and impact on survival of subcutaneous calcinosis in 86 patients with immune-mediated rheumatic diseases (IMRD).
I would like to raise the following concerns.
1.
Many statistical analyses and p-values are presented in the main body of the text, but there are no accompanying tables provided, making it challenging for readers to easily comprehend the key findings of the study.
For example, lines 158-160 discuss the calculation of odds ratios, the calculation of correlation and association analysis between different IMRD and subcutaneous calcinosis in section 3.2, as well as the calculation in section 3.3.2 for Outcomes and Survival, and so on.
2.
Tables 1 to 3 present descriptive statistical analyses; it is recommended that these tables incorporate analytical statistical analyses and provide information such as effect size, p-values, and other relevant details.
Author Response
1.
Many statistical analyses and p-values are presented in the main body of the text, but there are no accompanying tables provided, making it challenging for readers to easily comprehend the key findings of the study.
For example, lines 158-160 discuss the calculation of odds ratios, the calculation of correlation and association analysis between different IMRD and subcutaneous calcinosis in section 3.2, as well as the calculation in section 3.3.2 for Outcomes and Survival, and so on.
Answer: Thank you for bringing this to our attention! To enhance the clarity of the study’s key findings, we have included additional details regarding patient characteristics in Table 1. Moreover, we have replaced Tables 2 and 3 with Figures 1 and 2 for better visualization. Additionally, Tables 2,3 and 4 have been added to illustrate patient characteristics associated with calcinosis burden in the studied cohort.
2.
Tables 1 to 3 present descriptive statistical analyses; it is recommended that these tables incorporate analytical statistical analyses and provide information such as effect size, p-values, and other relevant details.
Answer: Incorporating p-values into Table 1, we used Kruskal-Wallis test for comparing different organ involvement. Tables 2 and 3 have been transformed into Figures 1 and 2, respectively.

Reviewer 2 Report
Comments and Suggestions for Authors
This manuscript described the relationship between calcinosis and clinical characteristics in rheumatic diseases.
There is one thing I would like to recommend; the results of multiple regression analysis do not appear in the results session while being mentioned only in discussion part instead. Please show the multiple regression result in table and explain it in the results part.
Author Response
There is one thing I would like to recommend; the results of multiple regression analysis do not appear in the results session while being mentioned only in discussion part instead. Please show the multiple regression result in table and explain it in the results part.
Answer: Thank you for pointing this out! Accordingly, we have included Table 2,3 and 4 to improve clarity and provide additional details for readers.

Reviewer 3 Report
Comments and Suggestions for Authors
I find the article interesting and well written. It is not a novel topic but the cohort is extense and varied.
Author Response
I find the article interesting and well written. It is not a novel topic but the cohort is extense and varied.
Answer: Thank you for your thoughtful review! We appreciate your recognition of the study’s topic and significance. Our decision to undertake this study was drive by the need to address existing limitations in diagnostic, monitoring, and targeted therapeutic approaches for calcinosis in patients with immune-rheumatic diseases. Despite being considered a “non-lethal” manifestation, subcutaneous calcinosis can profoundly impact patients’ quality of life and functionality. We aim to shed light on this aspect and contribute to improving the efficacy of treatment options for this condition.

Round 2
Reviewer 1 Report
Comments and Suggestions for Authors
No further comment